# A Fast Two-Stage Bilateral Filter Using Constant Time *O*(1) Histogram Generation

**DOI:** 10.3390/s22030926

**Published:** 2022-01-25

**Authors:** Sheng-Wei Cheng, Yi-Ting Lin, Yan-Tsung Peng

**Affiliations:** Department of Computer Science, National Chengchi University, Taipei City 11605, Taiwan; 108753502@g.nccu.edu.tw (S.-W.C.); 08360681@me.mcu.edu.tw (Y.-T.L.)

**Keywords:** image smoothing, gaussian filtering, bilateral filtering, *O*(1) complexity

## Abstract

Bilateral Filtering (BF) is an effective edge-preserving smoothing technique in image processing. However, an inherent problem of BF for image denoising is that it is challenging to differentiate image noise and details with the range kernel, thus often preserving both noise and edges in denoising. This letter proposes a novel Dual-Histogram BF (DHBF) method that exploits an edge-preserving noise-reduced guidance image to compute the range kernel, removing isolated noisy pixels for better denoising results. Furthermore, we approximate the spatial kernel using mean filtering based on column histogram construction to achieve constant-time filtering regardless of the kernel radius’ size and achieve better smoothing. Experimental results on multiple benchmark datasets for denoising show that the proposed DHBF outperforms other state-of-the-art BF methods.

## 1. Introduction

Edge preservation and smoothing of the content in an image is a basic problem in machine vision, computer graphics, and image processing. To remove shadows from a single input image, Yang et al. recovered a 3-D intrinsic image using edge-aware smoothing from the 2-D intrinsic image [1]. Stereo matching resolved ambiguities caused by nearby pixels with different parallaxes but similar colors using adaptive support weight-based trilateral filtering [2]. The salient regions of an image were extracted for image segmentation by the re-blurring model with bilateral and morphological filtering [3]. The joint bilateral filter was modified to up-sample the depth image with the high-resolution edge guidance for image super-resolution [4]. The guided image filter converted the structures of the guidance image to the output one so as to be available for transmission map refinement in image dehazing [5]. Yin et al. investigated the global, local, and social characteristics widely used in the image smoothing to develop the image reconstruction model [6]. Licciardo et al. implemented the edge-preserving image smoothing hardware architecture for real-time 60 fps tone mapping in 1920×1080 pixels of image [7]. The work [8] extracted fine details from underexposed or overexposed images using content-adaptive bilateral filtering in the gradient domain for multi-exposure image fusion. The bright-pass bilateral filtering was presented to estimate the scene illumination for low-light image enhancement [9].

Gaussian low-pass filtering is a technique that reduces pixels’ differences by weighted averaging for image smoothing in various applications. However, such low-pass filtering cannot preserve image details, e.g., edges and textures. The linear translation-variant function *f* then describes the above filtering process below:(1)$$f\left(p\right)=\sum_{q}K_{p,q}\left(Q\right)P_{q},$$
where $K_{p,q}$ denotes each pixel *q* centred at pixel *p* in the filter kernel *K*, and Q and P are guidance and input images, respectively. For example, the kernel of Bilateral Filtering (BF) [10] described by Equation (Equation 1) is formulated below:(2)$$K_{p,q}\left(Q\right)=\frac{1}{n}\exp\left(-\frac{{∥p-q∥}^{2}}{\sigma_{s}^{2}}\right)\exp\left(-\frac{{∥P_{p}-Q_{q}∥}^{2}}{\sigma_{r}^{2}}\right),$$
where *n* is a normalization factor, and $\sigma_{s}$ and $\sigma_{r}$ are the window size of the neighborhood expansion and the change of the edge amplitude intensity, respectively. Exponential distribution function is generally used in Equation (Equation 2) to calculate the influence of different spatial distances by $\exp(-\frac{{∥p-q∥}^{2}}{\sigma_{s}^{2}})$, and $\exp(-\frac{{∥P_{p}-Q_{q}∥}^{2}}{\sigma_{r}^{2}})$ describing the contribution of the pixel intensity range. When Q and P are identical, Equation (Equation 2) is simplified as a single image smoothing form.

In this paper, we extend the existing BF frameworks by proposing a novel Dual-Histogram BF (DHBF) method for better denoising results. The main contributions of ours are threefold.

We improve the range kernel in BF based on an edge-preserving noise-reduced guidance image, where isolated noisy pixels are removed to avoid erroneously judging those noisy pixels as edges;We adopt mean filtering based on column histogram construction to approximate the spatial kernel, achieving constant-time filtering regardless of the kernel radius’ size and better smoothing;We conducted an extensive experiment on multiple benchmark datasets for denoising and demonstrated that the proposed DHBF performs favorably against other state-of-the-art BF methods.

The remainder of this paper is organized in the following sections. Section 2 reviews the state-of-the-art BF methods. Section 3 describes the proposed method. Section 4 provides both the qualitative and quantitative evaluations, and runtime estimation. Finally, Section 5 draws the conclusions of this paper.

## 2. Related Works

Based on geometric similarity and photometric distance, BF involving the non-iterative, local, and simple spatial and range kernels achieved edge preservation and image smoothing at the same time [10]. This algorithm avoided ghosting artifacts along edges and textures by adapting an image with the spatial and range kernels in contrast to Gaussian filtering, which tends to blur an image globally. As the nature of brute force calculation for each pixel in BF, early research focused on reducing the time consumption. Computational intelligence (CI) is a synonym of soft computing, expressed for computers optimizing a specific task from data or experimental observation, while the common definition of CI has not yet existed in the literature [11,12,13]. According to our survey, there are several types of CI-based BF methods improving either computation time or image quality. In the following paragraphs, we further discuss BF-based methods in classic versus CI-based aspects

### 2.1. Classic Methods

In general, the previous BF studies can be categorized as histogram-based [14,15], quantization-based [16,17], and transformation-based BF [18,19,20,21,22,23,24,25] algorithms. To keep the constant time computation of BF, Porikli implemented the integral histogram database, avoiding the collection of redundant data in the spatial domain [14]. He et al. provided high accuracy of local window matching around color edges for image smoothing by injected locality-sensitive histogram to linear-time BF [15]. Based on the spatial decomposition, a uniform framework was presented to approximate the brute-force BF using several constant time spatial filters [16]. Yu et al. found the run time trade-off property in [16] to establish a speedup model based on the adaptive control of adjusting block size [17]. In contrast with the spatial domain filtering via histogram or quantization-based techniques, most BF algorithms applied the transformation process to high-dimensional convolutions to improve the computation performance and image quality [18,19,20,21,22,23,24,25].

### 2.2. CI-Based Methods

Chaudhury et al. used trigonometric range kernels to approximate the standard spatial Gaussian kernels of BF [18], which was done by the generalization of polynomial kernels in [14]. The shiftability of trigonometric kernels was further identified as a non-linear shiftable kernel simplifying the moving sum of a stack in image filtering, which was also available to parallel computation [19]. Sugimoto and Kamata proposed a compressive Fourier-based BF method to achieve a period length of Gaussian approximation [20]. Similarly, a pointwise-convergent Fourier series was applied to ensure the accuracy of filtering sub-pixels [21]. Optimized Fourier BF (OFBF) [22] was proposed by approximating the truncated Gaussian kernel. The deviation of the OFBF depends on the iteration parameters, including sinusoid numbers and sinusoid coefficients. OFBF uses an approximation model based on least-square fitting for a fixed period coefficient optimization. A sum of exponential functions approximated the range kernel of the BF to achieve O(1) complexity [26]. Based on the polynomial approximation of local histograms, an adaptive BF was proposed to achieve significant accelerations using only a few convolutions [23]. Guo et al. used a raised cosine function to approximate the range kernel of the BF to improve computation performance [27]. Similarly, the sparse approximation was conducted to optimize the cosine approximation of range kernels with spatial convolutions of BF [24].

When BF-based methods were used in denoising, image noises cannot be differentiated from details with its range kernel once noisy pixels have large differences from their neighborhood ones, retaining most noise in the denoising result. Gaussian-Adaptive BF (GABF) [25] then calculated the range kernel differently. In GABF, Gaussian filtering is applied to the input image to reduce noisy pixels before computing the range kernel. Using a noise-reduced smooth image as guidance, GABF can alleviate the above problem. However, adopting the range kernel calculated based on the smooth guidance image causes image details to be filtered out, thus degrading denoising quality. The two-pass filtering scheme adopted in GABF also requires an additional computation cost.

## 3. Proposed Method

We propose a novel Dual-Histogram BF (DHBF) method to achieve better denoising and edge-preserving smoothing with O(1) time complexity regardless of the kernel radius. It contains two stages as shown in Figure 1. In Stage 1, we produce the noise-reduced guidance map G using the proposed computationally inexpensive edge-preserving BF to remove noise but preserve image details. Next, Stage 2 outputs the final filtered image based on the proposed edge-preserving BF with the range kernel computed using the guidance image obtained from the output patch of Stage 1.

### 3.1. Local Histogram Generation

Many local histogram generation algorithms have been presented in previous studies [28,29,30,31,32]. The simplest method directly visited each pixel in the sliding window. The major bottleneck is the time-consuming search operation. Previous work [28] proposed to reduce the spatial redundancy in the calculation of histograms. There is a large intersection area between two consecutive pixels. Hence, the spatial redundancy was available to the incoming region while modifying the boundary information [28]. However, this algorithm consumes O(r) per pixel cost, where *r* is the search radius.

Inspired from the integral image approach, a constant time O(1) method computed the local histogram in a Cartesian data space constructing a superset of the cumulative image formulation, named integral histogram [29]. Based on the distributive characteristics, the sliding window can be divided into disjoint column regions to update histograms by vector-based operations [30]. This distributive property is applied to several image applications and histogram-based functions, including but not limited to entropy, distance norm, and cumulative distribution [31]. Peng et al. proposed a simple but effective differential histogram data structure [32] being applied to further improve the computational performance of the column histogram-based algorithm in [30].

Peng et al. have demonstrated that their proposed algorithm was more efficient than other O(1) local histogram generation algorithms [32]. Hence, we implement the differential column histogram structure in local histogram generation for both Stages 1 and 2 in the DHBF.

### 3.2. Two-Stage Filtering

Let I be the input image and G be the guidance image. For each incoming pixel $I_{x}$, the corresponding output pixel $J_{x}$ is calculated as:(3)$$J_{x}=\left(\sum_{s\in \Omega_{x}}K_{x,s}\left(I_{x},G_{s}\right)I_{x}\right)/\left(\sum_{s\in \Omega_{x}}K_{x,s}\left(I_{x},G_{s}\right)\right),$$
where $\Omega_{x}$ is the sliding window centred at *x*. $K_{x},s$ stands for the composite bilateral kernel, including the spatial kernel α and range kernel β, is given as: (4)$$K_{x,s}\left(I_{x},G_{s}\right)=\alpha \left(x,s\right)\beta \left(I_{x},G_{s}\right),$$
where α and β are formulated by the following two Gaussian-based exponential functions: (5)$$\alpha \left(x,s\right)=\exp\left(-\frac{{∥x-s∥}^{2}}{\sigma^{2}}\right),$$
(6)$$\beta \left(I_{x},G_{s}\right)=\exp\left(-\frac{{∥I_{x}-G_{s}∥}^{2}}{\delta^{2}}\right),$$
where σ and δ are the standard deviations for the spatial and range kernels. Replacing Equation (Equation 5) with a constant coefficient, we can simplify Equation (Equation 3) by the histogram-based computation expressed below: (7)$$J_{x}=\left(\sum_{l=l_{\min}}^{l_{\max}}\beta \left(I_{x},l\right)H_{x}\left(l\right)l\right)/\left(\sum_{l=l_{\min}}^{l_{\max}}\beta \left(I_{x},l\right)H_{x}\left(l\right)\right),$$
where $\left[l_{\min},l_{\max}\right]$ is the pixel intensity range, and $H_{x}$ is the local histogram corresponding to the sliding window $\Omega_{x}$ centered at *x* of the guidance image G.

To keep the center pixel’s origin intensity, we replace $G_{x}$ with $I_{x}$ in $H_{x}$. We adopt column histogram construction [32] to achieve O(1) time complexity regardless of the radius size of the kernel. In Stage 1, we generate the guidance image G by substituting $I_{x}$ for $G_{x}$ in Equation (Equation 7).

## 4. Experimental Results

As shown in Figure 2, Figure 3, Figure 4, Figure 5 and Figure 6, we chose five benchmark datasets, BSDS100 [33], Set5 [34], Set14 [35], Urban100 [36], and USC-SIPI [37] containing test images with a wide variety of contents. These images are converted to their grayscale versions for testing. We compare the proposed DHBF with three preivous BF methods, including BF [10], OFBF [22], and GABF [25]. These methods were all implemented with Matlab R2018b and tested in a laptop with Intel i7 CPU at 2.8 GHz and 16 GB RAM. We set both σ and δ to 15 in all compared methods. For DHBF, we set δ=45 in Stage 1 and δ=15 in Stage 2. The other parameters were set as the compared methods. To simulate noisy images, we add a random Gaussian noise with the standard deviation of 0.10.

To show the visual perception effects of edge-preserving image smoothing from each BF-based method, we selected one representative sample from each of the above five databases. We observed that some obvious noise artifacts still remain in the image after the BF [10], OFBF [22], and GABF [25] are applied to it. In contrast, our DHBF method further enhanced the output quality of the BF method based on the proposed two-stage framework. Our proposed DHBF can focus on the photometric representation between pixels to refine the image details regarding BF behavior. For example, our DHBF method further preserved the edges with noise reduction at the man head, butterfly wings, girl face, building, and texture chart shown in Figure 7, Figure 8, Figure 9, Figure 10 and Figure 11, respectively.

In addition to the above qualitative analysis, we also used peak Signal-to-Noise Ratio (PSNR) [38], Structural Similarity (SSIM) [39], Feature Similarity (FSIM) [40], and Gradient Magnitude Similarity Deviation (GMSD) [41] to perform Full-reference Image Quality Assessment (FR-IQA) of the output effects of different methods. PSNR is the ratio of the maximum possible power of a signal to the power of destructive noise that affects its accuracy. It is an evaluation index that can be used to quantify image distortion [38]. Given a ground-truth image and distortion image, SSIM measures the similarity of the two images to reflect the judgment of human eyes on image quality more appropriately than PSNR [39]. As the human visual system perceives images based on some low-level features, FSIM used phase consistency to describe the local image structure and extracted the gradient magnitude to supplement image variable attributes [40]. To explore the sensitive gradient changes of image distortion, GMSD computed the pixel-wise gradient magnitude map predicting perceptual image quality accurately [41].

Table 1 lists the FR-IQA results obtained using all the compared methods. We can see that DHBF performs favorably against other BF methods. Figure 12 demonstrate a runtime comparison of the compared BF methods with various kernel radius, ranging from 10 to 80 for processing 1024×1024 test images. As shown, both OFBF and DHBF methods run with constant time regardless of the radius sizes of the filter kernel. By contrast, BF and GABF require additional computation costs when a larger radius is used. According to all experiments mentioned above, DHBF performs the best for denoising while achieving O(1) time complexity.

## 5. Conclusions

This work proposed a novel Dual-histogram Bilateral Filtering (DHBF) with the range kernel computed based on the computationally inexpensive edge-preserving guidance image to produce better denoising results than existing state-of-the-art BF methods. In contrast to the compared computational-intelligence-based BF approach, OFBF, approximating the truncated Gaussian kernel, we calculated the spatial kernel using mean filtering based on column histogram construction, both achieving O(1) time complexity regardless of the kernel radius’s size. Subjective and objective experimental results on five benchmark datasets have verified that our proposed method performed favorably against other BF methods for edge-preserving smoothing.

## Figures and Tables

**Figure 1 sensors-22-00926-f001:**
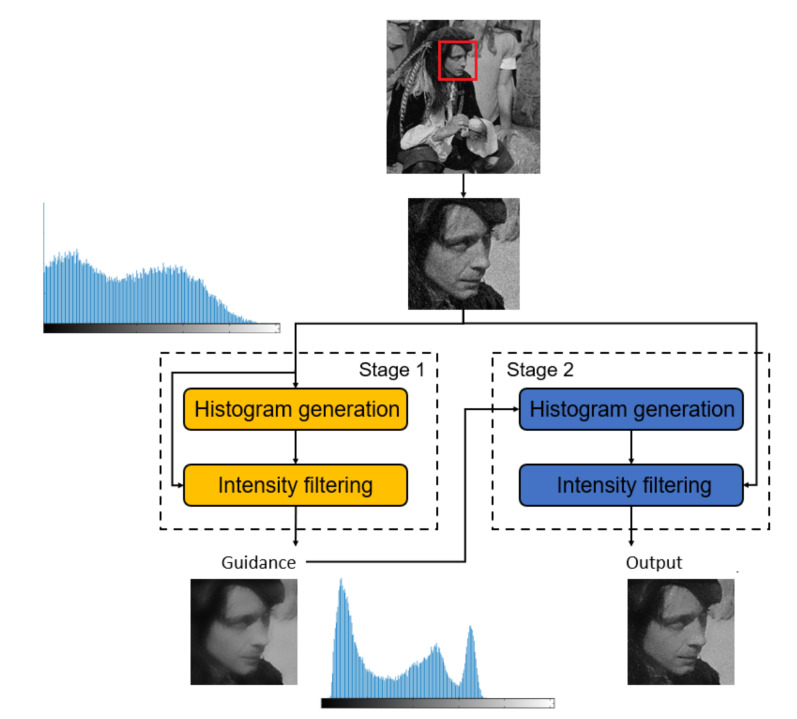
The flowchart of the DHBF method.

**Figure 2 sensors-22-00926-f002:**
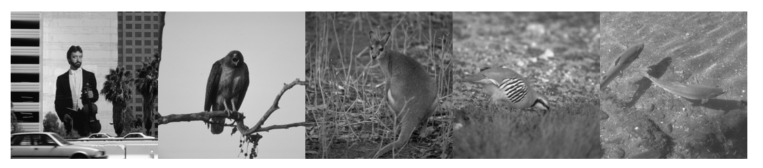
Thumbnails of BSDS100.

**Figure 3 sensors-22-00926-f003:**
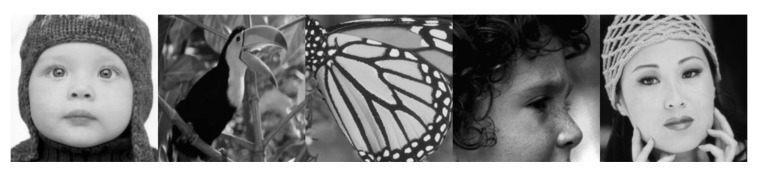
Thumbnails of Set5.

**Figure 4 sensors-22-00926-f004:**
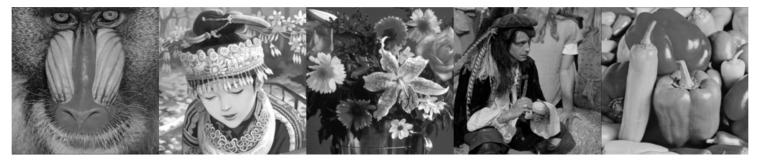
Thumbnails of Set14.

**Figure 5 sensors-22-00926-f005:**
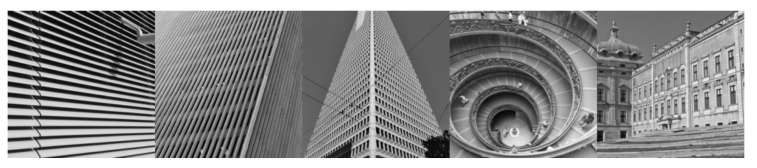
Thumbnails of Urban100.

**Figure 6 sensors-22-00926-f006:**
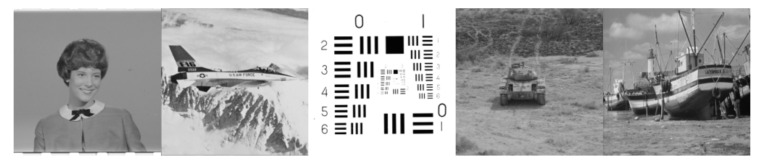
Thumbnails of USC-SIPI.

**Figure 7 sensors-22-00926-f007:**
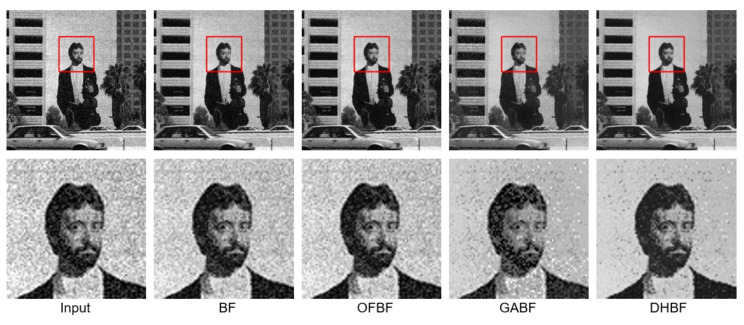
Visual comparison of the representative sample for image smoothing on BSDS100.

**Figure 8 sensors-22-00926-f008:**
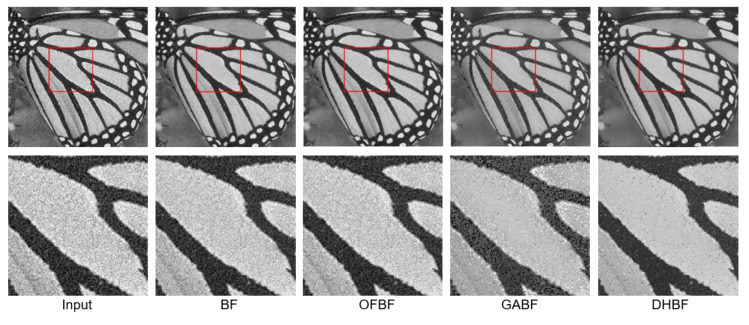
Visual comparison of the representative sample for image smoothing on Set5.

**Figure 9 sensors-22-00926-f009:**
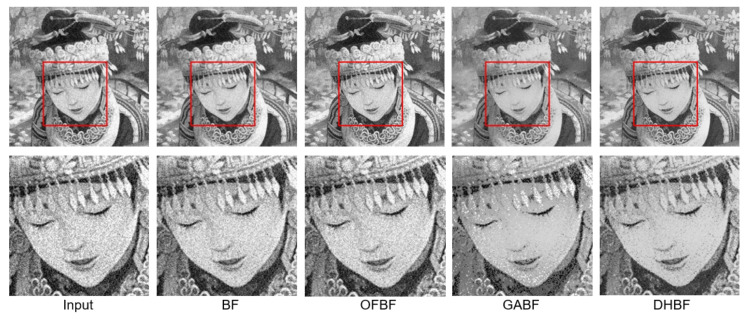
Visual comparison of the representative sample for image smoothing on Set14.

**Figure 10 sensors-22-00926-f010:**
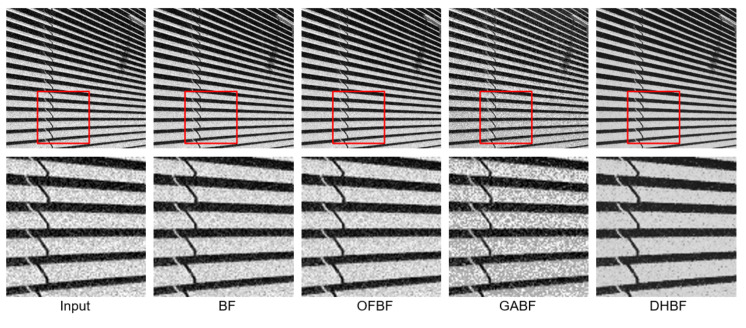
Visual comparison of the representative sample for image smoothing on Urban100.

**Figure 11 sensors-22-00926-f011:**
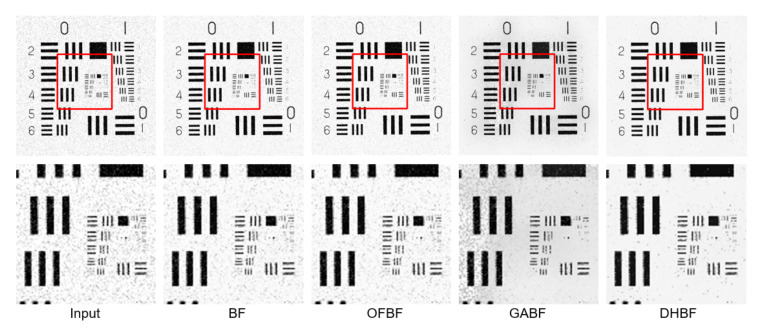
Visual comparison of the representative sample for image smoothing on USC-SIPI.

**Figure 12 sensors-22-00926-f012:**
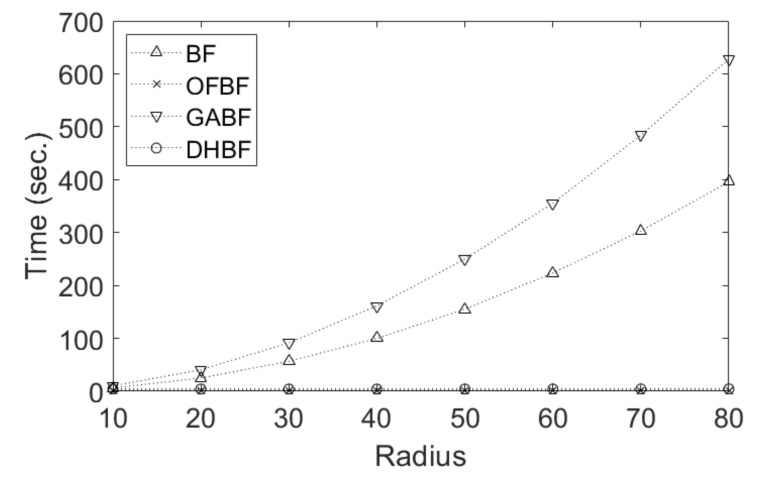
Runtime comparisons of the compared BF methods.

**Table 1 sensors-22-00926-t001:** Full-reference Image Quality Assessment for different methods. The best scores are in bold.

Dataset	Metrics	Methods			
		BF [10]	OFBF [22]	GABF [25]	DHBF
BSDS100 [33]	PSNR↑ [38]	21.8311	21.7637	22.4537	**23.6162**
	SSIM↑ [39]	0.4358	0.4335	0.5255	**0.5601**
	FSIM↑ [40]	0.7500	0.7479	0.7612	**0.8123**
	GMSD↓ [41]	0.1296	0.1306	0.1300	**0.1075**
Set5 [34]	PSNR↑ [38]	22.1226	22.0191	22.7614	**24.1221**
	SSIM↑ [39]	0.3349	0.3311	0.4531	**0.4859**
	FSIM↑ [40]	0.8400	0.8375	0.8436	**0.8817**
	GMSD↓ [41]	0.1367	0.1384	0.1326	**0.1071**
Set14 [35]	PSNR↑ [38]	21.8147	21.7265	21.9705	**23.3298**
	SSIM↑ [39]	0.4333	0.4302	0.5017	**0.5373**
	FSIM↑ [40]	0.8444	0.8421	0.8265	**0.8695**
	GMSD↓ [41]	0.1304	0.1318	0.1330	**0.1107**
Urban100 [36]	PSNR↑ [38]	21.6105	21.5481	20.8885	**22.7292**
	SSIM↑ [39]	0.5506	0.5484	0.5739	**0.6284**
	FSIM↑ [40]	0.8122	0.8107	0.7824	**0.8455**
	GMSD↓ [41]	0.1293	0.1305	0.1331	**0.1057**
USC-SIPI [37]	PSNR↑ [38]	22.1064	22.0483	22.9213	**24.4111**
	SSIM↑ [39]	0.3826	0.3806	0.5019	**0.5364**
	FSIM↑ [40]	0.8154	0.8140	0.8243	**0.8631**
	GMSD↓ [41]	0.1408	0.1416	0.1271	**0.1068**

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
