# Peer review of "A Fast Two-Stage Bilateral Filter Using Constant Time *O*(1) Histogram Generation"

_sensors, 2022, doi:10.3390/s22030926_

Round 1
Reviewer 1 Report
Bilateral Filtering (BF) is an effective edge-preserving smoothing technique with several applications in image processing. The presented paper introduces a novel Dual-Histogram BF (DHBF) method that exploits an edge-preserving noise-reduced guidance image to compute the range kernel, removing isolated noisy pixels for better denoising results. The method approximates the spatial kernel using mean filtering based on column histogram construction to achieve constant-time filtering. Experimental results on multiple benchmark datasets for denoising show that the proposed DHBF outperforms other state-of-the-art BF methods.
The work is well-written and structured. The bibliography used is abundant and up-to-date. The results are analyzed in correspondence with standard metrics for this type of task and consequently, the conclusions are supported by experimental evidence.
In order to improve the quality of the proposed work, this reviewer has minor concerns/remarks related below.
- It is required to improve the Introduction section in order to state the problem statement, novelty and contributions, and its relation to Sensors' scope. Also, I would like to recommend presenting the structure of the article in order to guide the readers.
- I recommend to separate the overview of the state-of-the-art presented in the Introduction as an independent section. e.g. 2. Related Works.
- The proposed method is presented as 'a fast two-Stage bilateral filter' and in Section 3 (line 153) refer that 'According to all experiments mentioned above, DHBF performs the best for denoising while achieving O(1) time complexity'. However, the authors did not show results on the computing time of the proposed method compared to the state-of-the-art.
- The experiments consider three previous BF methods for comparisons, including BF [9], OFBF [18], and GABF [21]. Why these methods were considered by the authors? I recommend including other approaches focused on the time complexity, both in the review of the related works and time comparisons, such as:
+ Gavaskar, R. G.; Kunal N. Chaudhury, K. N. Fast adaptive bilateral filtering. IEEE Trans. Image Process. 2019, 28, 779–790. DOI: 10.1109/TIP.2018.2871597
+ Dai, L.; Tang, L.; Tang, J. Speed up bilateral filtering via sparse approximation on a learned cosine dictionary. IEEE Trans. Circuits Syst. Video Technol. 2020, 30, 603–617. DOI: 10.1109/TCSVT.2019.2893322.
+ Zhang, X., & Dai, L. (2019). Fast bilateral filtering. Electronics Letters, 55(5), 258-260.
+ Guo, J., Chen, C., Xiang, S., Ou, Y., & Li, B. (2019). A fast bilateral filtering algorithm based on rising cosine function. Neural Computing and Applications, 31(9), 5097-5108.
+ Ghosh, S., & Chaudhury, K. N. (2019, September). Fast bright-pass bilateral filtering for low-light enhancement. In 2019 IEEE International Conference on Image Processing (ICIP) (pp. 205-209). IEEE.
+ among other relevant works on the subject matter.
- Could be interesting to know/study how the proposed method performs against other filtering methods not based on the bilateral filter.
Author Response
Please find the attached file to download our reply letter.

Reviewer 2 Report
It is well known that the bilateral filter can introduce several types of image artifacts: Staircase effect (intensity plateaus that lead to images appearing like cartoons) or Gradient reversal (introduction of false edges in the image). To deal with these artifacts the guided filter [5] has been proposed.
The proposed two-stage BF method also uses a guidance image to remove noises effectively while preserving edge structure.
#1. I’d like to ask the authors to specify the contribution of this study more differentiable from existing methods and why such study is important.
#2. The proposed method computes the guidance image in the first stage. I’d like to ask the authors to describe the guidance image in a mathematical formula.
#3. In general, the guided filtering method often suffers from texture-preserving. Therefore, I’d like to ask the authors how their method works for texture-preserving.
Author Response

(The authors gave the same response as above.)

Round 2
Reviewer 2 Report
The questions raised in the previous review are well answered in the revised manuscript.
Author Response
Thank you very much for helping improve the quality of our manuscript.